# Analyzing the Effects of Heat Treatment on SMAW Duplex Stainless Steel Weld Overlays

**DOI:** 10.3390/ma15051833

**Published:** 2022-03-01

**Authors:** Bernard-Maxmillan Sim, Sai-Hong Tang, Moath Alrifaey, Edwin-Nyon Tchan Jong

**Affiliations:** 1Faculty of Engineering, University Putra Malaysia, Serdang 43400, Malaysia; eng.alrifaiy2005@gmail.com; 2Department of Mechanical Engineering, Faculty of Engineering Technology and Built Environment, UCSI University, Taman Connaught, Kuala Lumpur 56000, Malaysia; 3Faculty of Engineering and Science, Curtin University of Malaysia, Miri 98000, Malaysia; edwinjong@curtin.edu.my

**Keywords:** duplex stainless steel, dilution, chemical composition, solidification, microstructure and microhardness

## Abstract

Duplex stainless steel (DSS) has a reasonably high resistance to chloride stress corrosion cracking for offshore and marine applications. However, DSS weld overlay has not been successfully demonstrated due to some inherent problems in achieving pitting and crevice corrosion resistance. In this research work, isothermal heat treatments (350, 650 and 1050 °C) with and different cooling rates have been performed DMR249 Grade A by using shield metal arc welding (SMAW) with an E2209 electrode. Micrographs have shown two phase microstructures of the DSS weld metal, the amounts of austenite phase increased with increment of post-weld heat treatment (PWHT) temperatures. The dilution has maintained consistent values except solution annealing that has shown the disappearance of the heat affected zone in micrographs. The weld metal hardness values increased with PWHT temperatures and remained low at solid solution annealing temperatures. The major alloying elements (C, Mo, Cr, Ni, N, and Fe) were analyzed, as these elements can contribute to intermetallic phases. The results showed that C and Cr content slightly increased with PWHT except for solid solution annealing, Mo showed consistently low content due to dilution effects. Ni maintained higher content, although the heat-treated samples showed slight fluctuations. Nitrogen produced consistent values, as recommended to prevent critical involvement in nitride precipitation.

## 1. Introduction

In the petrochemical and oil & gas industries, most corrosion-resistant alloy weld overlays are clad with austenitic stainless steel or high nickel alloys. Duplex stainless steel (DSS) offers better economics for its better resistance of pitting corrosion and the ability to maintain good mechanical properties as compared to other stainless steels [1]. DSS weld overlays have not been successfully demonstrated due to some inherent problems to achieve resistance to pitting and crevice corrosion caused by the intermetallic precipitations in the heat-affected zone. DSS is typically manufactured in annealed conditions because it has a high content of alloying elements, such as chromium, molybdenum, and nitrogen. Their structures are generally consisted of austenite (γ) (face-centered cubic structure) and ferrite (body-centered cubic structure) [2]. An alternative method to improve the material’s pitting corrosion resistance with reliable and affordable cost is using corrosion resistant alloy (CRA) cladding, whiech has been demonstrated to be more cost effective, considering the higher strength, better long-term reliability and light weight [3].

CRA weld overlays are formed through a process in which one or more metals are fused or joined to form a complete corrosion-resistant alloy protective layer on the surface of the substrate material. This weld overlay is chemically and/or metallurgically bonded to the base metal with a consistent composition alloying element interface between the corrosion-resistant alloy and the carbon steel substrate [3]. The term weld overlay designates the application of a relative thickness layer with at least 3 mm as recommended by industry practitioners. The corrosion resistance of weld deposits is strongly dependent on the amount and nature of the chemical elements and heat cycles. The shield metal arc welding (SMAW) process is the most common and is a basic welding process being recognized in all industries. This welding process is relatively simple, low cost and adaptable to any confined space and can be carried out remotely. Heat treatment for DSS weld overlay on low alloy carbon steel substrate has not presented in any industry welding standards. Most of the industrial welding codes only cover for individual groups of material [4,5]. Consequently, the behavior and material properties of CRA weld overlays are less understood as compared to other steels. In particular, the effect of PWHT on DSS weld overlays and the low alloyed carbon steel substrates under hydrogen (350 °C), stress relief (650 °C), and solution annealing (1050 °C) are unknown. The aim of this paper is to determine the effect of post-heating on the chemical dilution, microhardness, and microstructure of a DSS E2209 weld overlay.

## 2. Materials and Methods

A medium-strength low-alloy carbon steel DMR 249 Grade A steel plate with dimensions of 450 mm × 450 mm × 20 mm was used in this research study. This material is commonly used in a wide range of general industries due to its strength, good weldability, low cost, and high availability. Welding consumable E2209-16 @ 4.0 mm was selected to join standard DSS materials, as it offers excellent corrosion resistance to pitting and stress cracking in aqueous chloride applications and offshore environments [6]. Table 1 lists the chemical elements, the materials, and the welding consumable used in this study. A flat welding position was selected because simple and straight forward, the limitation of position produces deeper penetration due to gravity acceleration and it has the significant impact to the characteristic to weld motel pool and output of geometry. The welding parameter process shown in Table 2 was carefully studied to minimize the secondary phases in order to improve pitting corrosion resistance and maintain good bonding strength. Heat input with moderate cooling rates is recommended to ensure that the austenite phase formation is sufficient to achieve corrosion-resistant properties. As reported by Maria et al., higher heat input would result in very slow cooling rates that were favored in the precipitation of sigma phase as evidenced by X-ray powder diffraction (XRD) [7].

### 2.1. Solidification and Ferrite Prediction

A traditional Schaeffler diagram showing the constitution of the DSS weld metal was plotted. This diagram assists to predict the type of microstructures in weld metals and parent materials. Chromium equivalent (Cr_eq_) is defined as the ferrite former or stabilizer, whereas nickel equivalent (Ni_eq_) is described as the austenite former or stabilizer. The solidification of primary austenite can occur when the ratio of Cr_eq_ to Ni_eq_ is below 1.25. Liquefaction cracking is believed to occur with the critical partitioning of impurities, such as sulfur and phosphorus [8,9]. According to Chakrabarti et al. [10], if the ratio of Cr_eq_ to Ni_eq_ is higher than 1.95, the solidification mode will be fully ferritic due to the strong diffusivity of chromium and molybdenum in the ferrite phase. The solidification and transformation sequence of the DSS microstructure is “L → L + δ → δ → δ + γ.” The Schaeffer diagram can quantitatively forecast the microstructure in the weld metal based on the Cr_eq_-to-Ni_eq_ ratio [11]. Equations (1) and (2) are used to obtain the weighting factors:

Austenite Stabilizing Element:(1)$$\mathrm{Nickel}\;{\mathrm{Ni}}_{\mathrm{eq}}\;=\;\%\mathrm{Ni}\;+\;30\left(\%C\right)\;+\;0.5\left(\%\mathrm{Mn}\right)$$

Ferrite Stabilizing Element:(2)$$\mathrm{Chromium}\;{\mathrm{Cr}}_{\mathrm{eq}}\;=\;\%\mathrm{Cr}\;+\;\%\mathrm{Mo}\;+\;1.5\left(\%\mathrm{Si}\right)\;+\;0.5\left(\%\mathrm{Nb}\right)$$

### 2.2. Heat Treatment

The PWHT process parameters involve three fundamental steps, which are heating, soaking, and cooling. The heating temperature from ambient to 315 °C was considered an unrestricted parameter. Above 315 °C, heating was carried out using heating elements at a rate no greater than 335 °C per hour divided by the maximum metal thickness in mm. During the holding period, the differential soaking temperatures were maintained within ±20 °C upper and lower limits. The soaking time of PWHT was governed by wall-thickness, most of industrial code of standard requested for one (1) hour per 25 mm, except for solid solution annealing temperature (1050 °C), the soaking period was maintained at 2 h. After soaking was completed, the cooling process was carried out at a rate no greater than 335 °C per hour divided by the maximum metal thickness of the thicker part in mm until 315 °C. Air cooling test samples could only be removed from insulation blanket when cooling temperature has reached to 315 °C whereas water quenching test samples were immediately quenched with running water at 33 ± 2 °C. Table 3 presented the information of soaking temperatures and cooling methods of test samples.

### 2.3. Macro Hardness

The macro-hardness test ensures that the material quality and mechanical properties meet the standard requirements after surface treatment. In this study, Vickers hardness was used to measure the resistance of the material to indentation. The Vickers hardness test gives accurate measurements for ceramic materials, alloyed steels, and weld metals. In this test, the material surface is indented using standard loads applied for a specified length of time. The diagonal length of the impression is proportional to the driving force of the pyramid slope area of the indentation. The pyramid has a standard square base and a pyramidal indenter with an impression angle of 136 °C. A load indenter at 10 kgf is released and slowly applied onto the test specimen surface, without any vibration, at a temperature of (25 ± 5) °C. The load is held up for 10–15 s before it automatically returns to its original (or neutral) position. The average of two mean readings is then taken to calculate the hardness value. The two indentation diagonal lengths should be limited to not more than 5% of the flat surface. At least three readings must be taken—from the weld metal, the heat-affected zone, and the base material [12].

### 2.4. Microstructure Charaterization

Metallographic examination and physical property analysis were carried out based on the ASTM A923–Method A [13]. A quick analysis of the DSS microstructure detrimental phase was done to determine the resistance of the material. Moreover, the detrimental phase of the weld overlay was investigated using a scanning electron microscope (SEM). Plane grinding was done using silicon carbide with a reduction abrasive grit size 240 until 3 µm was achieved in the final polishing process [14]. The etching process for the metallographic test samples was electro etching done in 10% NaOH solution per 100 g of the sample using. The distilled water was changed from 1–3 V dc for 15 s to generate better differences in the austenite and ferrite phases. The etched microstructure was metallurgically evaluated with a light microscope (ML) and a field emission scanning electron microscope (FE-SEM), both of which give a high-quality image of the interface between the weld overlay, the base material, the secondary phase, the retained austenite (light phase), and the ferrite (in the dark phase) [13].

### 2.5. Analysis of Chemical Dilution

The DSS-CRA weld overlay works well with low-alloy or high-tensile carbon steel depending on the extent of dilution [15]. In cladding or weld overlay applications, the dilution effect needs to be kept at a minimum to maintain the original alloy elements and to ensure the cladding retains its corrosion and wear resistance [16]. The concept of dilution is shown in the schematic illustration of Figure 1. The weld overlay alloying elements in the fusion zone are affected by the concentration and the dilution of chemical elements in the area of the weld metal ($A_{\mathrm{fm}}$) and the base metal area ($A_{\mathrm{bm}}$).

Dilution (D) can be measured based on the amount of substrate material mixed with the weld metal in the fusion zone. In most applications, it is crucial to control the chemical dilution between the two materials to produce the required microstructure and properties for the intended service. The metallographic microstructure of the transition zone can be evaluated using a high magnification optical light microscope. The morphological microanalysis was done using FE-SEM, and the chemical alloying elements were determined using a SPECTROLAB metal analyzer (Kleve, Germany). The weld bead dilution indicating the chemical composition was estimated at 30%. The methodology of calculation for the first layer weld deposit was based on 70% welding bead and 30% base metal. The chemical composition for the dilution of the second layer remained the same, at 30%. The resulting chemistry of the second layer deposit (30% of 30%) was 9% base metal and (100%–9%) or 91% welding bead. The estimated specific chemical elements of each weld layer are presented in Table 4.

## 3. Results and Discussion

### 3.1. The Effect of PWHT on Mode of Solification

The solidification mode and transformation weld overlay performed under different heat treatment conditions were studied. The austenite grain precipitate in the solid-state was determined from the nucleation and growth along the ferrite boundaries. The amount of austenite and ferrite precipitation in the DSS weld deposits strongly depends on the amount of chemical composition and the number of heat cycles. The solidification modes can be determined by using chromium equivalent (Cr_eq_) and nickel equivalent (Ni_eq_) values as shown in Table 5. In the current experimental work, the solidification mode has been categorized in ferrite mode with austenite and ferrite microstructures at the ratio of Cr_eq_/Ni_eq_ in the range between 2.24 and 2.33 with red highlighted as indicated in Figure 2.

### 3.2. Microstructure of Weld Deposits

In this research work, isothermal heat treatments in the range 350–1050 °C for various holding times and different cooling rates have been applied on duplex stainless steel weld overlays to study the kinetic of alpha embrittlement, carbide, chromium nitride, chi and sigma phase precipitation. Figure 3a–i show the micrograph of cross sections of the weld metals which present an austenite-ferrite phase microstructure. Sample No. 1 (as welded), shown in Figure 3a, revealed two phase microstructures in the austenite-ferritic matrix, where it can be observed that the amount of austenite phase formation was greater that of ferrite phase which was indicated by a higher pitting corrosion resistance. Samples No. 2 and 3 (Figure 3b,c) had undergone PWHT at 350 °C and indicated that the grain boundary austenite (GBA) and widmanstatten austenite (WA) had slightly decreased. Samples No. 4 and 5 showed that the volume fraction of ferrite was slightly deceased in the weld metal as shown in Figure 3d,e, which might be due to lots of precipitated ferrite phase that was dissolved into the gamma (γ) phase. By increasing the solution anneal temperature to 1050 °C, numerous intra-granular austenite grains structures were revealed, whereas the amount ferrite phases were further reduced. The micrographs in Figure 3f,i depict that austenite grains did not show any favorable growth direction. The indications have shown that the entire contents of austenite in weld metal and heat affected zone are increased after conducting solution annealing treatment followed by rapid quenching in water in order to avoid the sensitizing temperature zone varying from 800 °C to 500 °C.

### 3.3. Weld Metal Dilution

The alloying elements of the DSS weld overlay were greatly influenced by the mixing of the filler and the base metal compositions, especially the dilution between the two dissimilar metals (i.e., DSS and carbon steel). Several factors should be considered when determining dilution effects, such as heat input, polarity, cooling rates, electrode size, electrode stick-out, type of shielding, and bead spacing. All these variables will influence the dilution of the weld overlay. The dilution of weld metal chemical composition was estimated using a 30% approach, as confirmed by Jagesvar et al. [15].

In this research study, the locations of actual chemical data are obtained at 3 mm from the fusion line to indicate/determine the alloying element segregation/migration. The geometrical plotting method was employed to estimate the area of dilution between the two materials. Figure 4 shows the actual dilutions obtained in the range of 30.98% and 33.14% for as-weld and heat treated samples nos. 1 to 5. The samples nos. 6 to 9, have undergone high temperature heat treatment at 1050 °C with rapid water quenching and their micrographs showed the disappearance of the heat affected zone therefore dilution was invisible.

### 3.4. Chemical Elements Distribution

The partitioning of the major chemicals elements such as carbon, chromium, nickel, molybdenum and nitrogen which can contribute intermetallic phases (sigma, chi, nitrides, carbides etc.) in conventional duplex stainless steels have been measured for various post weld heat treatment and solid solution annealing temperatures.

#### 3.4.1. Carbon (C)

Carbon is the main chemical element to provide strength for austenite grain structures, particularly at elevated temperatures and to prevent solidification cracking. Although carbon is an austenite-former, this chemical element is limited to the lowest practicable levels to prevent the possible formation of M_23_C_6_ carbides in the ferrite phase or at the grain boundaries. The carbon content in the weld metals as shown in Figure 5a indicated higher values than predicted, which might due to the carbon migration from the base metal to the weld metal during the heat input cycle. As-welded values ranged from 0.030 wt.% to 0.034 wt.%. After PWHT at 350 °C, the carbon alloying element slightly increased from 0.031 wt.% to 0.036 wt.%. PWHT at 650 °C indicated quite extensive carbon migration from the base metal, with the results showing a significant increased from 0.032 to 0.038 wt.%. Carbon has a high affinity to form carbide precipitates with chromium between the grain boundaries in the fusion zone. Excessive carbon content can introduce undesirable precipitates within the ferrite phase, such as carbides (M_7_C_3_/M_23_C_6_).

#### 3.4.2. Chromium (Cr)

This is a main chemical component that confers pitting corrosion resistance protection to steels with a minimum content of 20 wt.%. It also promotes ferrite phase formation and increases the strength in solid solution, however an extremely high content can result in low ductility and inferior toughness. The as-welded sample showed chromium contents ranging from 20.33 wt.% to 24.67 wt.%. PWHT samples No. 2 to 5 showed a slightly high chromium content in between 20.34 wt.% and 24.87 wt.%, respectively, because of the higher temperature of the heat treatment and the aging time that could cause the chromium content in the Fe-rich phase to progressively increase during the ferrite decomposition. However, the selected PWHT samples No. 6 to 9 were subjected to a reheating process at solid solution temperature (1050 °C) with 2-h soaking time and immediately quenched with water. The results showed consistently low chromium content at 19.88 wt.% and 23.68 wt.% as indicated in Figure 5b, which was due to the solid-state phase transformation from austenite to alpha (α) ferrite, the higher volume fraction of ferrite prevents the chromium and molybdenum from spreading over a greater region and becoming diluted. It has also been reported by Guo et al. [17] that the content of chromium decreased in the ferrite phase and slightly improved in the austenite phase, with the same solution annealing temperature.

#### 3.4.3. Molybdenum (Mo)

Molybdenum is usually added as a secondary alloying element, especially to improve pitting corrosion resistance in chloride environments and strengthen the steel at elevated temperature. The minimum content of molybdenum shall be controlled within 4 wt.% higher amount of molybdenum will cause detrimental due to formation of intermetallic precipitates. The test results as shown in Figure 5c have indicated the contents are in between 2.17 wt.% and 3.19 wt.%, which approximately represents a 10% to 38% reduction from the original electrode chemical composition (3.53 wt.%) due to dilution effects. Overall molybdenum has shown consistent results, except for samples No. 4 and 5 which underwent stress relief heat treatments and have shown a small increment in the lower range values. According to Kai and Sie [18], a higher molybdenum content has a greater potential to form the sigma and chi phase in DSS as a result of post-weld heat treatment.

#### 3.4.4. Nickel

This element is commonly added into DSS to promote formation of the austenite phase matrix and improve the fracture toughness at lower temperatures. The results showed consistent lower values, as indicated in Figure 5d, in between 8.42 wt.% and 8.81 wt.%, although the heat-treated samples showed slight fluctuations. The nickel content in the weld metal was reduced from 15%–19% of its original value of 10.44 wt.%, which is possibly due to the dilution effect of the welding heat input as well as the heat treatment process, which promotes the formation of secondary austenite. Nickel is a good solution strengthener to improve impact toughness for all kinds of stainless steels; it reduces the ductile to brittle fracture transition temperature (DBTT).

#### 3.4.5. Nitrogen (N)

Nitrogen’s primary function is to improve and stabilize the formation of the austenite phase. It is normally added in DSS and controlled within the range of 0.08 and 0.35 wt.%. The as-welded samples indicated values of 0.16 wt.% and 0.18 wt.%. PWHT 350 °C, samples No. 2 and 3 showed the lowest nitrogen content at 0.16 wt.% and 0.17 wt.%, whereas, samples No. 4 and 5 subjected to PWHT 650 °C revealed similar values as welded sample. The highest obtained values of nitrogen were from 0.16 wt.% to 0.19 wt.% for the solid solution annealing samples No. 6 to 9 which enhance the corrosion properties as referred in Figure 5e.

#### 3.4.6. Ferrite (Fe) Count

Most of oil and gas industries have stated the average ferrite content should be in between 40–60% by volume for the weld metal and heat affected zone regions for duplex stainless steel [19]. DSS weld overlays with local heat treatments have therefore not been critically studied in such details [20]. In this research study, the results had revealed that ferrite count determination in the region of weld metal overlay were decreased with high temperatures, as referred in Figure 5f. The welded sample indicated ferrite contents of 18.8 ± 2.14 vol.% PWHT 350 °C with air cooling resulted in a slightly higher ferrite content at 20 ± 1.78 vol.% as compared to the water quenching technique, which produced a marginally lower ferrite count 19.4 ± 2.05 vol.%. Similarly, with PWHT 650 °C air cooling also resulted in a lower ferrite content of 17.5 ± 1.57 vol.%, whereas the water-quenching technique showed a reduction in ferrite value of 16.3 ± 1.34 vol.%, due to the fact slow cooling rates are more favorable to austenite formation due to the sufficient time for nucleation growth. However, the amount of ferrite in the weld metals decreased considerably from 10.6 ± 1.36 vol.% to 11.9 ± 0.82 vol.% the increment of temperature to solution annealing at 1050 °C. If appropriate time were given for austenite formation, an optimal equilibrium fraction for the weld area could be achieved. Figure 3 shows the ferrite content in the area of the weld metal overlay under the influence of different heat treatments.

### 3.5. Macro-Hardness

The macro-hardness measurement of the weld overlay cross-section profile ass shown in Figure 6. The weld profile can be categorized into three characteristic regions: (a) weld metal zone (E2209-16); (b) heat affected zone (HAZ); and (c) base metal region (DMR 249A). The macro hardness of the DSS weld overlay was obtained in the range between 193 Hv and 241 Hv, as displayed in Figure 7. Likewise, HAZ showed lower macro-hardness values between 197 Hv and 241Hv while the base metal indicated a higher macro-hardness, ranging from 204 Hv to 376 Hv, due to the different heat treat temperatures and cooling effects applied. The hardness of the first layer of the treated and non-treated weld overlay test samples was slightly higher than that of the 2nd layer (weld cap). This slight increase in hardness could be due to heat treatment, which caused carbide coarsening and precipitation. The weld metal hardness values increased after being subjected to PWHT at 350 °C and 650 °C because of the refining of the grains and the slight increment in the ferrite volume fractions, which were detected in the zone of fusion, as highlighted in Figure 5f. The weld metal for samples 6 to 9 was subjected to a solution annealing temperature of 1050 °C with water quenching. The test results showed consistent low hardness values in the range of (192–204) Hv at the weld overlay and HAZ regions. However, the hardness values substantially increased from 364 Hv to 376 Hv at the base metal, with this higher hardness value appearing to have larger grains and a noticeably equiaxed ferritic structure as well as the presence of a large amount of martensite.

## 4. Conclusions

In this research work, isothermal heat treatments in the range 350–1050 °C with various holding times and different cooling rates have been performed on DSS weld overlays on carbon steel substrates. The solidification mode was categorized as ferrite mode with austenite and ferrite microstructures displaying Cr_eq_/Ni_eq_ ratios in between 2.24 and 2.33.

Micrographs (as welded samples) showed two phase microstructures of the DSS weld metal, as compared to post-weld heat treatment samples, where the austenite phase formation is greater than that of ferrite phase. Generally, after PWHT at 350 °C, the microstructures showed a slight reduction in austenite phase. After PWHT at 650 °C, micrographs showed a further reduction in the volume fraction of ferrite due to precipitation of ferrite phase dissolved into gamma phase. A solution annealing temperature of 1050 °C produced numerous intra-granular austenite grains structures and the amount ferrite phases was significantly reduced.

Based on industrial practice, SMAW process dilution is taken as 30%. The produced dilution of the DSS weld overlay was 32.87% for as-welded sample, a slightly increment of about 1% for PWHT at 350 °C and a small reduction of approximately 2% was observed for PWHT at 650 °C. Solution annealing temperature at 1050 °C with water quenching revealed the disappearance of the heat affected zone in micrographs.

A slight increment in hardness for the first layer was observed as compared to the 2nd layer of the weld deposits and the weld metal hardness values increased after conducting PWHT. The solid solution annealing produced consistently low hardness values at the WM and HAZ and was unfavorable for base metal and the hardness increased substantially and produced larger grains, noticeable in the equiaxed ferritic structure, together with the occurrence of a large quantity of martensite.

The major alloying elements (C, Mo, Cr, Ni, N, and Fe) were analyzed, as these elements can contribute to the formation of intermetallic phases. The results showed that C and Cr content slightly increased as a result of PWHT, except for solid solution annealing, Mo showed consistently low content due to dilution effects. Ni maintained a higher content, although the heat-treated samples showed slight fluctuations. Nitrogen produced consistent values as recommended to prevent critical nitride precipitation.

## Figures and Tables

**Figure 1 materials-15-01833-f001:**
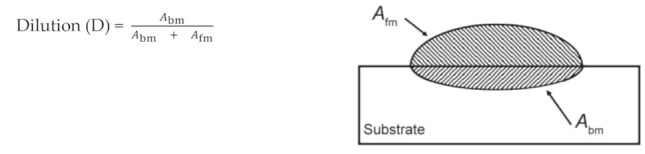
Illustration of dilution of base metal and filler metal in fusion weld.

**Figure 2 materials-15-01833-f002:**
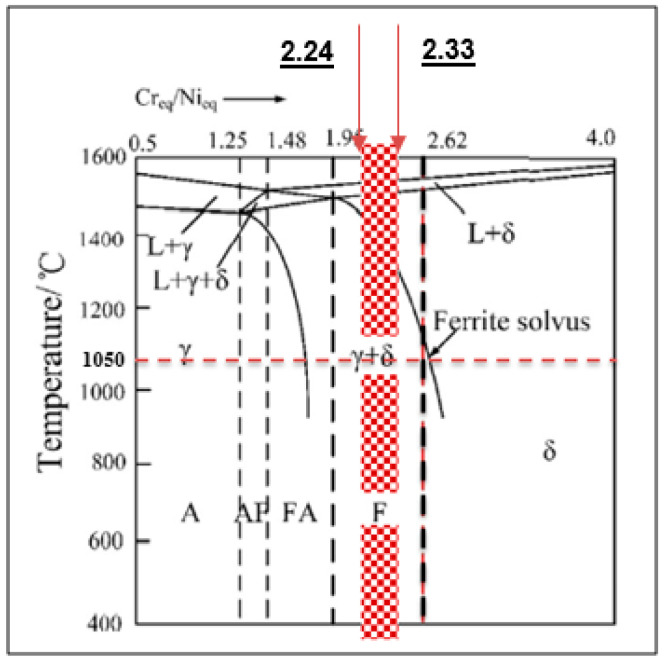
The DSSs solidification modes in the ternary Fe-Cr-Ni Section.

**Figure 3 materials-15-01833-f003:**
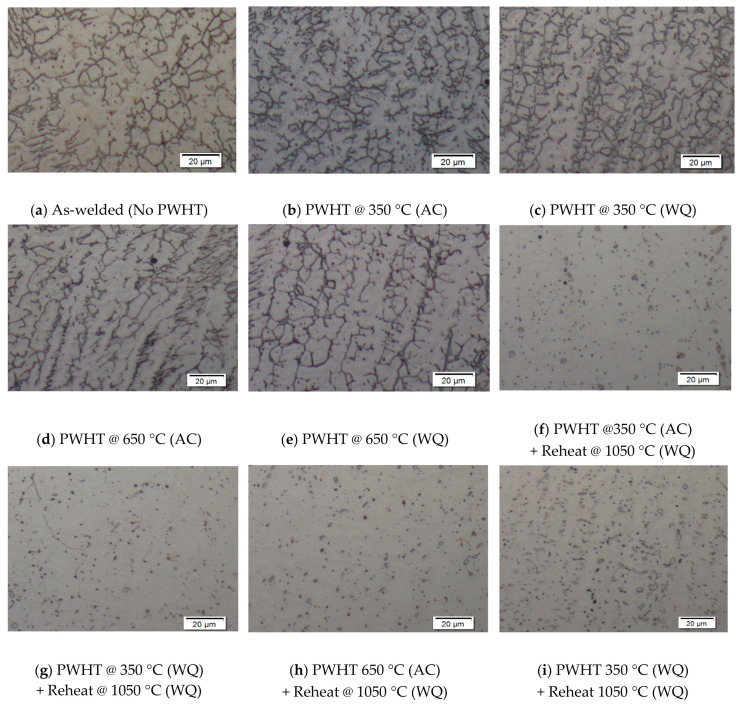
SEM Image of at fusion zone with three different heat treatments (Dark phase ferrite, white phase-austenite).

**Figure 4 materials-15-01833-f004:**
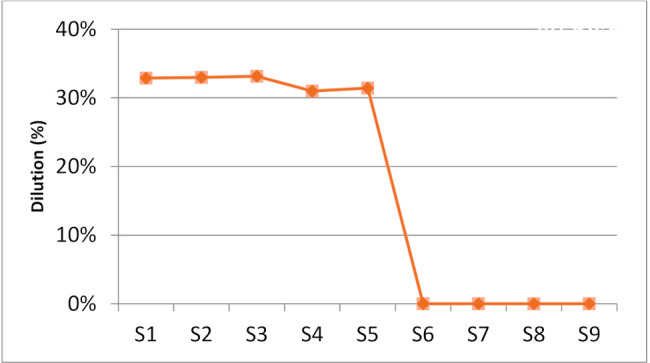
Comparison of dilution with the effect of heat treatment processes.

**Figure 5 materials-15-01833-f005:**
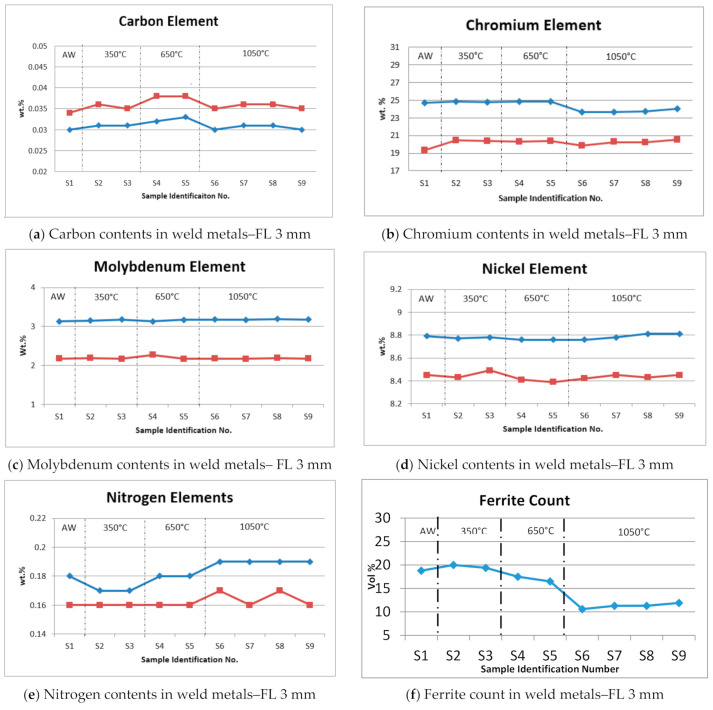
Chemical elements and ferrite counts for as-welded and heat treated samples.

**Figure 6 materials-15-01833-f006:**
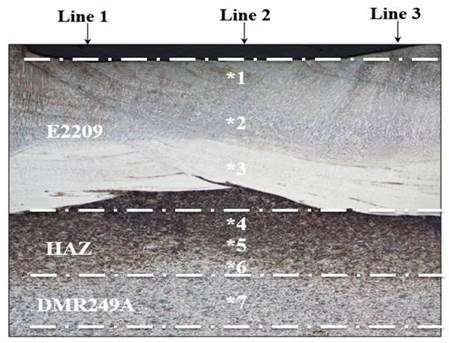
Micro-hardness test location at the cross-section.

**Figure 7 materials-15-01833-f007:**
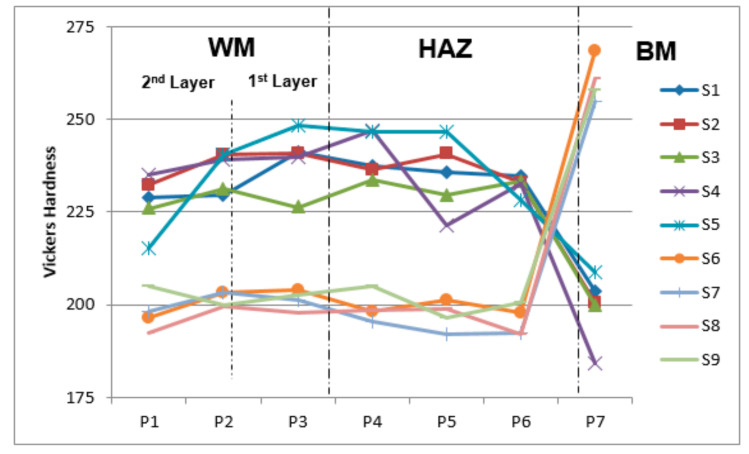
Comparison of macro-hardness in the cross-section area.

**Table 1 materials-15-01833-t001:** Chemical elements for base metal and consumable welding.

Materials	Element of Alloy (wt.%)									
	Cr	C	Mo	Ni	Mn	Ti	Nb	Cu	Si	N
Electrode, E2209-16 @ 4 mm	23.49	0.014	3.55	10.45	1.32	0.028	0.012	0.051	0.53	0.16
Base Metal, DMR-249 Grade A	0.30	0.110	0.05	1.05	1.65	0.06	0.05	0.30	0.40	−

**Table 2 materials-15-01833-t002:** Welding parameter for welding consumable.

Filler		Range of Process Parameter				
Type	Dia. (mm)	Voltage/Current	Amperes (A)	Voltages (V)	Travel Speeds (cm/min)	Heat Inputs (KJ/cm)
E2209-16	4.0	DC–RP	120.0–125.0	22.0–25.0	13.0–14.5	12.2–12.9

**Table 3 materials-15-01833-t003:** Post weld heat treatment conditions.

Sample ID	Temperatures and Cooling Methods
S1	As-welded (without heat treatment)
S2	at 350 °C with air cool
S3	at 350 °C with water quench
S4	at 350 °C with air cool
S5	at 350 °C with water quench
S6	at 350 °C with air cool and reheat at 1050 °C with water quench
S7	at 350 °C with water quench and reheat at 1050 °C with water quench
S8	at 650 °C with air cool and reheat at 1050 °C with water quench
S9	at 650 °C with water quench and reheat at 1050 °C with water quench

**Table 4 materials-15-01833-t004:** Prediction dilution for alloying elements in each weld layer.

Weld Metal	Dilution	C	Ni	N	Mn	Cr	Mo	Si
1st Layer	30.0%	0.044	7.62	0.14	1.42	16.47	2.49	0.49
2nd Layer	30.0%	0.026	9.60	0.17	1.35	21.32	3.22	0.52

**Table 5 materials-15-01833-t005:** Shows the solidification mode and microstructure of weld overlay.

Specimen	Cr_eq_	Ni_eq_	Cr_eq_/Ni_eq_	Solidification Mode	Microstructure
1	22.51	9.93	2.27	Ferrite	A + F
2	22.63	9.88	2.29	Ferrite	A + F
3	22.55	9.99	2.26	Ferrite	A + F
4	22.61	9.92	2.28	Ferrite	A + F
5	22.73	9.86	2.31	Ferrite	A + F
6	22.19	9.89	2.24	Ferrite	A + F
7	22.47	9.95	2.26	Ferrite	A + F
8	23.19	9.96	2.33	Ferrite	A + F
9	22.76	9.96	2.29	Ferrite	A + F

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
