# Peer review of "Analyzing the Effects of Heat Treatment on SMAW Duplex Stainless Steel Weld Overlays"

_materials, 2022, doi:10.3390/ma15051833_

Round 1

Reviewer 1 Report

This paper concerns the effect of post heat-treatment on the duplex stainless steel overlay structure and concludes that the microstructure (and properties) can be tuned by adjusting the welding and heat treatment process.  This research is very comprehensively done and the paper is reasonably well written.  There are two minor concerns, though, and they are:

1) editorial:  there are many grammatical errors or unfinished sentences.  They need to be fully corrected.  Also necessary is to review the "Introduction".  The first paragraph sounds odd and the last paragraph describing the structure of the paper is not necessary.  

2) technical:  the change in microstructure with heat-treatment is interesting and central to merit of this paper.  yet, it is somewhat difficult to follow, perhaps due to insufficient description of the expected microstructure at given temperature based on the phase diagram.  

Author Response

Reviewer No. 1

This paper concerns the effect of post heat-treatment on the duplex stainless steel overlay structure and concludes that the microstructure (and properties) can be tuned by adjusting the welding and heat treatment process.  This research is very comprehensively done and the paper is reasonably well written.  There are two minor concerns, though, and they are:

Editorial; there are many grammatical errors or unfinished sentences.  They need to be fully corrected.  Also necessary is to review the "Introduction".  The first paragraph sounds odd and the last paragraph describing the structure of the paper is not necessary.  

[Agreed, we have corrected the grammatical errors and added several sentences to address this, structure of the paper is removed.]

Technical; the change in microstructure with heat-treatment is interesting and central to merit of this paper.  Yet, it is somewhat difficult to follow, perhaps due to insufficient description of the expected microstructure at given temperature based on the phase diagram.

[Agreed, we have modified the text to make it clearer and added several sentences to provide more information.]

Reviewer 2 Report

Materials

Title: Analyzing Effects of Heat Treatment on SMAW Duplex Stainless Steel Weld Overlays

Manuscript #: 871969

In the paper " Analyzing Effects of Heat Treatment on SMAW Duplex Stainless Steel Weld Overlays ", the authors analyzed the effects of post-weld heat treatment on the chemical dilution, microhardness and microstructure of Shield Metal Arc Welding duplex stainless steel weld overlays. The influences of heat treatment temperature, together with the cooling methods were taken into consideration. The conclusion of this study could, to some extent, promote the application potentials of DSS weld overlays. Before this manuscript can be taken into further consideration, the authors should respond to the following comments/suggestions.

  1. In the abstract, the authors stated that “this result confirms that DSS can be used in multidisciplinary applications”, it is too arbitrary to give this statement.
  2. The authors should explain the rationality for choosing the location 3mm from the fusion line.
  3. Some typos, such as “in the heat-affected fusion zone [19&20]”, what is “heat-affected fusion zone”? and where is [19&20]?
  4. I could not understand “an extremely high heat input would result in very slow cooling rates and a high chance of intermetallic phase formation”.
  5. In 2.3 Heat Treatment, what do you mean by “there is no case for when the rate goes beyond 335C per hour”?
  6. In 2.3 Heat Treatment, some details, such as the heating rate, cooling rate, holding time, are not provided.
  7. What is “Figures 3(f) to (i)”?
  8. What is “SA” in table 3?
  9. In figure 6, what do the two colors represent?
  10. In figure 8, there are total 9 samples but only list “S1…S7”.
  11. The conclusion part should be revised. I couldn’t get the major points of this paper.
  12. I think the major issue for this paper is that the authors have published another similar paper entitled “the influence of Post Weld Heat Treatment Precipitation on Duplex Stainless Steels Weld Overlay towards Pitting Corrosion”. Some results, such as Figure 3 and Figure 5 are totally the same for the two papers.

Author Response

Reviewer No. 2

In the paper "Analyzing Effects of Heat Treatment on SMAW Duplex Stainless Steel Weld Overlays", the authors analyzed the effects of post-weld heat treatment on the chemical dilution, micro-hardness and microstructure of Shield Metal Arc Welding duplex stainless steel weld overlays. The influences of heat treatment temperature, together with the cooling methods were taken into consideration. The conclusion of this study could, to some extent, promote the application potentials of DSS weld overlays. Before this manuscript can be taken into further consideration, the authors should respond to the following comments/suggestions.

  1. In the abstract, the authors stated that “this result confirms that DSS can be used in multidisciplinary applications”, it is too arbitrary to give this statement.

[Agreed, we have corrected and added several sentences to address this.]

  1. The authors should explain the rationality for choosing the location 3mm from the fusion line.

      [We have added the information to address this, please see introduction and section 3.3.]

  1. Some typos, such as “in the heat-affected fusion zone [19&20]”, what is “heat-affected fusion zone”? and where is [19&20]?

[Agreed, typos have been removed.]

  1. I could not understand “an extremely high heat input would result in very slow cooling rates and a high chance of intermetallic phase formation”.

      [We have added the information to address this, please see section 2 (above table 1).]

  1. In 2.3 Heat Treatment, what do you mean by “there is no case for when the rate goes beyond 335C per hour”?

[We have modified the text to make it clearer. Please see section 2.3]

  1. In 2.3 Heat Treatment, some details, such as the heating rate, cooling rate, holding time, are not provided.

[We have modified the text to make it clearer and added several sentences to provide more information.  Please see section 2.3 and Table 3]

  1. What is “Figures 3(f) to (i)”?

      [We have revised to Figures 3 (vi) to (ix).]

  1. What is “SA” in table 3?

[SA = Solution Annealing, we have revised the text to “reheat” and make it clearer Please see section Table 3]

  1. In figure 6, what do the two colors represent?

[We have changed the Figure 6 to Figure 5. Red and blue colors indicated lower and upper values using SPECTROLAB metal analyzer.]

  1. In figure 8, there are total 9 samples but only list “S1…S7”.

[Agreed, we have changed the Figure 8 to Figure 7 with complete test samples from S1 to S9.]

  1. The conclusion part should be revised. I couldn’t get the major points of this paper.

[Agreed, we have revised and added several sentences to address this.]

  1. I think the major issue for this paper is that the authors have published another similar paper entitled “the influence of Post Weld Heat Treatment Precipitation on Duplex Stainless Steels Weld Overlay towards Pitting Corrosion”. Some results, such as Figure 3 and Figure 5 are totally the same for the two papers. –

[We have tried to eliminate duplication, Figure 5 was maintained to provide more information and linkage with other results, it allows readers to more quickly skim the text and decide what sections to read in more detail.]

Round 2

Reviewer 2 Report

I was satisfied with the response.